# Molecular Cloning and Functional Analysis of 1-Deoxy-D-Xylulose 5-Phosphate Reductoisomerase from *Santalum album*

**DOI:** 10.3390/genes12050626

**Published:** 2021-04-22

**Authors:** Yueya Zhang, Haifeng Yan, Yuan Li, Yuping Xiong, Meiyun Niu, Xinhua Zhang, Jaime A. Teixeira da Silva, Guohua Ma

**Affiliations:** 1Guangdong Provincial Key Laboratory of Applied Botany, South China Botanical Garden, Chinese Academy of Sciences, Guangzhou 510650, China; tulipazhyy@yahoo.com (Y.Z.); liyuan@scbg.ac.cn (Y.L.); 13838553436@163.com (Y.X.); niumeiyun@scbg.ac.cn (M.N.); xhzhang@scib.ac.cn (X.Z.); 2Computer Science Department, University of the Chinese Academy of Sciences, Beijing 100049, China; 3Sugarcane Research Institute, Guangxi Academy of Agricultural Sciences, Nanning 530007, China; gstsyhf@163.com; 4Independent Researcher, P.O. Box 7, Ikenobe 3011-2, Kagawa-ken 761-0799, Japan; jaimetex@yahoo.com

**Keywords:** 1-deoxy-D-xylulose 5-phosphate reductoisomerase, chlorophylls and carotenoids, MEP pathway, *Santalum album*, sandalwood sesquiterpenoids

## Abstract

Sandalwood (*Santalum album* L.) heartwood-derived essential oil contains a high content of sesquiterpenoids that are economically highly valued and widely used in the fragrance industry. Sesquiterpenoids are biosynthesized via the mevalonate acid and methylerythritol phosphate (MEP) pathways, which are also the sources of precursors for photosynthetic pigments. 1-deoxy-D-xylulose-5-phosphate reductoisomerase (DXR) is a secondary rate-limiting enzyme in the MEP pathway. In this paper, the 1416-bp open reading frame of *SaDXR* and its 897-bp promoter region, which contains putative conserved *cis*-elements involved in stress responsiveness (HSE and TC-rich repeats), hormone signaling (abscisic acid, gibberellin and salicylic acid) and light responsiveness, were cloned from 7-year-old *S. album* trees. A bioinformatics analysis suggested that *SaDXR* encodes a functional and conserved DXR protein. *SaDXR* was widely expressed in multiple tissues, including roots, twigs, stem sapwood, leaves, flowers, fruit and stem heartwood, displaying significantly higher levels in tissues with photosynthetic pigments, like twigs, leaves and flowers. *SaDXR* mRNA expression increased in etiolated seedlings exposed to light, and the content of chlorophylls and carotenoids was enhanced in all *35S::SaDXR* transgenic *Arabidopsis thaliana* lines, consistent with the *SaDXR* expression level. *SaDXR* was also stimulated by MeJA and H_2_O_2_ in seedling roots. α-Santalol content decreased in response to fosmidomycin, a DXR inhibitor. These results suggest that *SaDXR* plays an important role in the biosynthesis of photosynthetic pigments, shifting the flux to sandalwood-specific sesquiterpenoids.

## 1. Introduction

*Santalum album* is an economically important tropical woody species of the Santalaceae family that is well known for its aromatic heartwood and essential oil [1]. The sesquiterpenoid alcohols (*Z*)-α-santalol, (*Z*)-β-santalol, (*Z*)-epi-β-santalol and (*Z*)-α-exo-bergamotol, which are the main components of *S. album* essential oil, play a vital fragrance-defining role in the fragrance industry [2]. These sandalwood sesquiterpenoids also possess various biological properties [3]. The formation of heartwood has a decades-long cycle due to the slow growth of the trees, and this property, coupled with overexploitation, habitat destruction and plant disease, have threatened global sandalwood resources. Compared to the long cycle times associated with artificially planting sandalwood and then laboriously harvesting and processing the heartwood, or the expensive cost of chemical synthesis of sandalwood sesquiterpenoids, metabolic engineering is a more economic and efficient system [4,5]. For this reason, it is critical to elucidate the molecular mechanism underlying the biosynthesis of sandalwood terpenoids.

Although isoprenoids are structurally and functionally the most diverse group of plant metabolites, all isoprenoids are biosynthesized from two universal precursors, isopentenyl diphosphate (IPP, C5) and dimethylallyl diphosphate (DMAPP, C5), which are derived from two distinct pathways, namely the plastidial methylerythritol phosphate (MEP) pathway and the cytoplasmic mevalonate acid (MVA) pathway (Figure 1) [6,7]. All enzymes in the MEP pathway are encoded by nuclear genes, then imported into plastids [8,9]. Consequently, the MEP pathway drives the function of the plastids and principally provides IPP and DMAPP precursors for the biosynthesis of growth hormones (gibberellins, cytokinins and abscisic acid), terpenoids (monoterpenoids and diterpenoids) and photosynthetic pigments (chlorophylls and carotenoids). Sesquiterpenoids, triterpenoids and brassinosteroids are mainly produced in the cytoplasmic MVA pathway (Figure 1). The physical cellular separation of MEP and MVA pathways in plastids and cytosol, respectively, facilitates the optimal supply of precursors required in each cell compartment, but some research suggests that there is an exchange between plastidial MEP and cytoplasmic MVA pathways via cross-talking IPP [6,10].

To date, almost all the studies that examined the molecular mechanism of the biosynthesis of sandalwood sesquiterpenoids focused on the function of downstream terpene synthases (TPSs) and cytochrome P450-dependent monooxygenase [11,12,13,14,15,16,17]. Even though 1-deoxy-D-xylulose-5-phosphate reductoisomerase (DXR), which is a rate-limiting enzyme that transforms 1-deoxy-D-xylulose 5-phosphate (DXP) to 2-C-methyl-D-erythritol 4-phosphate (MEP), controls flux through the MEP pathway [18,19], there are few studies that have focused on the function of DXRs, and there are no reports in sandalwood. DXRs play an important role in enhancing the accumulation of isoprenoids, including secondary metabolites and plastidial terpenoids (Appendix A). For this reason, we studied DXR to better understand the biosynthesis of sesquiterpenoids and photosynthetic pigments (chlorophylls and carotenoids) in sandalwood.

In this study, the *SaDXR* gene was cloned based on the *S. album* transcriptome database [20]. The expression profiles of *SaDXR* in response to different treatments (methyl jasmonate (MeJA), hydrogen peroxide (H_2_O_2_) and light) and in various tissues, were assessed. Subcellular localization and promoter analysis of *SaDXR* were also performed. The relationship between *SaDXR* and the biosynthesis of chlorophylls, carotenoids, and sandalwood-specific sesquiterpenoids was illuminated by overexpressing *SaDXR* in *Arabidopsis thaliana* and assessing its response to a DXR inhibitor, fosmidomycin (FOS).

## 2. Materials and Methods

### 2.1. Plant Materials and Treatments

Young (7-year-old) *S. album* trees were grown in South China Botanical Garden (SCBG), in Guangzhou, China, under natural conditions (Appendix A). Separately, fully ripe *S. album* seeds were collected from 15-year-old trees growing in SCBG. Seeds were sown in compost in clay pots and placed in a phytotron (75% relative humidity (RH), 28 °C) and left in the dark for 40 days. The etiolated seedlings were exposed to light. Whole seedlings were collected at 0, 1, 3, 6, and 24 h after exposure to light, wrapped in tin foil, and stored immediately in liquid nitrogen. After the remaining seeds were germinated on a sand bed (20 cm deep), seedlings 10 cm in height were transferred to plastic pots (10 cm in diameter × 12 cm high) that were filled with a mixture of sand, peat, and soil (3:1:1, *v*/*v*/*v*), and maintained in a greenhouse between March and November. Seedlings were grown in natural sunlight with an average day/night temperature of 25/18 °C, 70–80% RH, together with the host plant, *Kuhnia rosmarnifolia* Vent [21]. More developed seedlings, 30 cm in height, were transferred to a phytotron for 3 weeks, with a 16-h photoperiod at 28/23 °C, 80 µmol m^−2^ s^−1^ and 75% RH [20]. Plant roots were gently washed and placed in distilled water for 6 days, and water was changed every 2 days prior to treatments. Roots of seedlings were dipped in a solution with a final concentration of 5 mM H_2_O_2_, 1 mM (MeJA) and 20 µM FOS. Roots were harvested at 0, 1, 3, 6, 9, 12, 24 h and 7 days after exposure to two elicitors (H_2_O_2_ and MeJA) and inhibitor (FOS), rapidly frozen in liquid nitrogen, and frozen at −80 °C until use.

*A. thaliana* ecotype Columbia (Col-0) plants were grown in plastic pots containing a mixture of topsoil and vermiculite (1:2, *v*/*v*), and the pots were placed in a phytotron at 22/20°C (day/night), 70% RH and a 16-h photoperiod (100 µmol m^−2^s^−1^) until needed for transformation and to determine the content of chlorophylls and carotenoids. Transgenic plants, derived as explained in 2.8, were grown on half-strength (macro- and micro-nutrients) Murashige and Skoog (MS) medium [22] with 0.8% (*w*/*v*) agar, 3% (*w*/*v*) sucrose, and 30 mg L^−1^ hygromycin B to screen overexpression lines. After about two weeks, in vitro *A. thaliana* plants were transferred to plastic pots.

### 2.2. Isolation Full-Length cDNA and Genomic DNA of SaDXR from S. album

Total RNA was extracted from mature leaves of 7-year-old *S. album* trees using Column Plant RNAout2.0 (Tiandz Inc., Beijing, China) according to the manufacturer’s instructions. cDNA was reverse transcribed with M-MLV Reverse Transcriptase (Promega, Madison, WI, USA). Based on the annotation of unigenes in our transcriptome database [20], the *SaDXR* unigene was isolated and used to design gene-specific primers for 5′ and 3′ rapid amplification of cDNA ends (RACE) (Appendix A). RACE reactions were performed with the SMARTer RACE cDNA Amplification Kit (Clontech Laboratories Inc., Mountain View, CA, USA). RACE products were cloned into the pMD18-T vector (Takara Bio Inc., Dalian, China) and sequenced. Full-length *SaDXR* was amplified by polymerase chain reaction (PCR) using primers that were designed based on sequences by combining the 5′-RACE and 3′-RACE fragments (Appendix A).

Total genomic DNA (gDNA) was isolated from mature leaves of 7-year-old *S. album* trees using the Column Plant DNAout kit (Tiandz, Inc.), and amplified with a pair of primers (gDNAF and gDNAR; Appendix A). The obtained genomic sequence was aligned to the full-length cDNA of *SaDXR* to confirm introns and exons.

### 2.3. Sequence, Protein Structure and Phylogenetics Analysis

The *SaDXR* open reading frame (ORF) was blasted in NCBI (https://blast.ncbi.nlm.nih.gov/Blast.cgi, accessed on 20 March 2019). The theoretical isoelectric point and molecular weight of the putative SaDXR protein were identified by ExPASy (https://www.expasy.org/, accessed on 20 March 2019). Amino acid sequences were aligned with DNAMAN version 7 and neighbor-joining phylogenetic trees were constructed using the bootstrap method on MEGA 5.1. The transmembrane domain of SaDXR was predicted by TMHMM version 2.0 (http://www.cbs.dtu.dk/services/TMHMM/, accessed on 20 March 2019). The 2D and 3D structures of SaDXR were determined by the Secondary Structure Prediction Method (SOPMA) (http://npsa-pbil.ibcp.fr/, accessed on 15 April 2019) and SWISS-MODEL (http://swissmodel.expasy.org, accessed on 15 April 2019), respectively. The gDNA sequence of *SaDXR* was compared to the gDNA of *Vitis vinifera DXR* (*VvDXR*) (Phytozome, accession no. GSVIVG01007752001), *Populus trichocarpa DXR* (*PtDXR*) (Phytozome, accession no. Potri.015G076200) and *A. thaliana DXR* (*AtDXR*) (Phytozome, accession no. AT5G62790).

### 2.4. Cloning of the SaDXR Promoter and Putative cis-Element Analysis

Fusion primer and nested integrated PCR (FPNI-PCR) [23,24] was used to isolate the upstream region of the *SaDXR* gene using gDNA as the template. The FPNI-PCR primers are listed in Appendix A. The putative transcription start site (TSS) of the cloned promoter was predicted by the TSSP program (http://linux1.softberry.com/berry.phtml?topic=tssp&group=programs&subgroup=promoter, accessed on 15 April 2019). PlantCARE software (http://bioinformatics.psb.ugent.be/webtools/plantcare/html/, accessed on 15 April 2019) was used to analyze the putative *cis*-elements in the *SaDXR* promoter.

### 2.5. Subcellular Localization of SaDXR

The *SaDXR* ORF without the termination codon was amplified by the KOD FX kit (Toyobo Inc., Shanghai, China) using primers YFP-Bgl II-F and YFP-Kpn I-R (Appendix A). The product was inserted into the *Bgl*II and *Kpn*I sites of the pSAT6-EYFP-N1 vector driven by the 35S Cauliflower mosaic virus (35S) promoter. *35S::SaDXR-YFP* and control *35S::YFP* were introduced into mesophyll protoplasts that were extracted from the leaves of 5-week-old *A. thaliana* plants by polyethylene glycol (PEG)-mediated transient transformation [25]. Transformed protoplasts were incubated in light for 20 h at 20 °C (see Section 2.1), and detected using a Zeiss LSM 510 confocal laser scanning microscope (Zeiss, Jena, Germany).

### 2.6. Quantitative Real-Time PCR (qRT-PCR) and Semi-Quantitative RT-PCR Analysis of SaDXR Gene Expression Profiles

For qRT-PCR and semi-quantitative RT-PCR, total RNA was extracted from the leaves of 4-week-old overexpression *A. thaliana* lines, different tissues (roots, twigs, stem sapwood, leaves, flowers, fruit and stem heartwood) of 7-year-old *S. album* trees (Appendix A), the roots of 8-month-old *S. album* seedlings exposed to two elicitors (H_2_O_2_ and MeJA), and etiolated seedlings from the light treatment. In addition, cDNA was prepared as described in Section 2.2. qRT-PCR was used to determine *SaDXR* expression profiles in *S. album* and overexpression *Arabidopsis* lines with the iTaq Universal SYBR Green Supermix (Bio-Rad Laboratories, Inc., Hercules, CA, USA) on an ABI 7500 Real-time system (ABI, Foster City, CA, USA). PCR consisted of initial denaturation at 95 °C for 2 min, 40 cycles of 95 °C for 15 s, 60 °C for 1 min, and plate reading after each cycle. *S. album Actin1* (NCBI accession no. HM232849.1) and *AtActin2* (NCBI accession no. AT3G18780) were used as reference genes. The relative expression level of genes was determined by the 2^−ΔΔCT^ method with three independent biological replicates [26]. Semi-quantitative RT-PCR used the Premix Taq (LA Taq version 2.0) (Takara Bio Inc., Shiga, Japan) in a 20 µL reaction mixture. The PCR conditions were 94 °C for 3 min followed by 30 cycles (27 cycles for *AtActin2*) of 94 °C for 30 s, 57 °C for 30 s, 72 °C for 1 min and 72 °C for 8 min. Finally, PCR products were verified by gel electrophoresis. All primers used are shown in Appendix A.

### 2.7. Metabolite Extraction and Gas Chromatography-Mass Spectrometry (GC/MS) Analysis

The ground dried roots (0.12 g) that were treated with FOS for 7 days were used to extract metabolites with 1 mL of hexane (Fisher Scientific, Waltham, MA, USA) in a GC vial by end-over-end mixing for 72 h at room temperature. The hexane phase without ground roots was obtained by centrifuging at 2000× *g* for 15 min at 4 °C, then hexane was evaporated to dryness. Finally, the residue was dissolved in 50 µL hexane and dodecane (0.014 mg ml^−1^; Shanghai Macklin Biochemical Co., Ltd., Shanghai, China), added as the internal standard for GC/MS analysis [17,27]. The relative content of sesquiterpenoids was identified using the internal standard and measurements were made from three independent biological replicates.

GC-MS was performed on a GC-2010 Gas Chromatograph (Shimadzu, Suzhou, China) equipped with a GCMS-QP2010 Plus Mass Spectrometer (Shimadzu). The mixtures of metabolites were analyzed by injecting 1 µL into an HP-5MS capillary column (30 m × 0.25 mm, film thickness 0.25 µm). Chromatographic conditions were as follows: injector 250 °C; ion source 250 °C; MS interface 250 °C; oven program from 50 to 250 °C at 5 °C min^−1^, then held for 2 min, then 5 °C min^−1^ to 300 °C, then held for 20 min; helium was used as the carrier gas at a flow rate of 1.0 mL min^−1^; scan range was 20–550 µm at a sampling rate of 2.0 scans/s. MS spectra were compared to NIST05, NIST05s and Wiley 9 library, and the compounds were identified based on positive identification.

### 2.8. Construction of Plant Overexpression Vector, Transformation and Regeneration of A. thaliana

*SaDXR* cDNA was inserted into the *Nco*I site of pCAMBIA1302 under the control of the 35S promoter by the In-Fusion HD Cloning Kit (Takara Bio Inc.). The pCAMBIA1302::35S:*SaDXR* plasmid was introduced into *Agrobacterium tumefaciens* EHA105, then transformed into *A. thaliana* Col-0 via the floral-dip method [28]. Putatively transformed *A. thaliana* lines were selected on half-strength MS medium supplemented with 30 mg L^−1^ hygromycin B. Homozygous T3 transgenic seedlings were identified by semi-quantitative RT-PCR.

### 2.9. Contents of Chlorophyll and Carotenoids in Transgenic A. thaliana Lines

Total chlorophyll and carotenoid contents were extracted in the dark with 5 mL acetone (80%) at 4 °C from ground leaf tissues (70 mg) of 5-week-old transgenic *A. thaliana* lines and wild type (WT) plants [29]. Total chlorophyll and carotenoid contents were determined using the methods described by Lichtenthaler [30], using a spectrophotometer (PERSEE, TU-1810, Shanghai, China) at 470, 645 and 662 nm.

### 2.10. Statistical Analysis

Statistical analyses were carried out using IBM SPSS 19.0 (IBM Corp., Armonk, NY, USA). All data represent the average of three biological replicates and standard error. Data was assessed by one-way analysis of variance (ANOVA) followed by Duncan’s multiple range test at *p* < 0.05 to denote significant differences between means.

## 3. Results

### 3.1. Molecular Cloning and Sequence Analysis of the Full Length SaDXR

The initial unigene of the candidate *S. album DXR*, which was identified within a transcriptome data set, had high homology with other plant *DXR*s [20]. Depending on the DNA fragment, gene-specific primers were designed for 3′- and 5′-RACE. Using 3′- and 5′-RACE, the full-length sequence of *SaDXR* was deduced by splicing the sequence of the initial unigene, and was then confirmed by PCR with specific primers and further sequencing. The full-length *SaDXR* sequence is 1826 bp long and contains the complete 1416 bp (Genbank accession no. MH018575) ORF plus 161 and 249 bp at the 5′ UTR and 3′ UTR, respectively. The ORF encodes a putative protein of 471 amino acids with a theoretical isoelectric point (pI) of 6.51 and calculated molecular weight of 51.02 kDa. The Protein-Blast against NCBI databases indicated that SaDXR had high homology with other plant species’ DXRs. SaDXR had the highest identity (87%) to the predicted DXR from *Vitis vinifera* (accession no. XP_002282761.1), and also had high similarity to the DXR of *Manihot esculenta* (86% identity, accession no. XP_021616674.1), *Hevea brasiliensis* (86% identity, accession no. XP_021669849.1), *Camellia sinensis* (86% identity, accession no. AKE33276.1), *Actinidia arguta* (85% identity, accession no. AID55340.1), *Betula platyphylla* (85% identity, accession no. AHX36947.1), *Prunus avium* (85% identity, accession no. XP_021826007.1), *Jatropha curcas* (84% identity, accession no. NP_001295720.1), and *Populus trichocarpa* (84% identity, accession no. XP_002318048.2) (Figure 2). There was a conserved NADPH binding site (GSTGSIGT) of a ketol acid reductoisomerase and a transit peptide for the plastid motif Cys-Ser-Ala/Met/Val/Thr near the N-terminal of *SaDXR* (Figure 2) [31,32].

### 3.2. Analysis of the *SaDXR* Gene

SaDXR gDNA (accession no. MH018577), 6226 bp long, possessed a complete coding region. The cDNA sequence was aligned with the gDNA. A comparison between SaDXR, VvDXR, PtDXR and AtDXR was carried out at the genomic level. The DXR of all four species had 12 exons and 11 introns, and the sequences of the exonic regions had 100% identity with the respective coding regions (Appendix A). The length and sequences of exons were similar in the four DXRs. Additionally, all intron boundaries followed an NGT-AGN pattern. The main difference between them lay in the introns, including the length and similarity of sequences. SaDXR and VvDXR had more similar genomic organization and sequences with higher homology than AtDXR and PtDXR, suggesting a relationship with the similarity of the deduced protein.

### 3.3. Protein Structure and Evolutionary Analysis of SaDXR

The secondary structure of SaDXR was predicted by SOPMA, which showed that the putative protein contains 168 random coils, 153 α-helices, 100 extended strands, and 50 β-turns. SaDXR was predicted to have a similar secondary structure as VvDXR, AtDXR and PtDXR (Appendix A). The 3D homology-based model of SaDXR had 42.75% sequence identity to *Plasmodium falciparum* DXR (PDB: 4gae) [33], and 82% of residues from SaDXR were modelled with PfDXR (Appendix A). There were also a few differences in the 3D structure between SaDXR and PfDXR (Appendix A). SaDXR had many common features with other DXR family members, which employ NADPH and Mn^2+^ (or Mg^2+^) as cofactors [33]. Specific amino acids involved in the binding with NADPH are illustrated in Appendix A.

A phylogenetic tree was constructed based on DXRs from other organisms, other than animals and fungi, which do not possess the MEP pathway. DXRs originated from an ancestral bacterial gene and evolved into four distinct clusters of bacteria, algae, gymnosperms and angiosperms. The evolutionary pattern of DXRs is consistent with the evolutionary relationship between lower and higher organisms. SaDXR lies in the DXR group with most other angiosperms and has highest homology with *V. vinifera* (Figure 3). The bioinformatics analyses strongly suggest that SaDXR is a plant DXR that possesses a functional DXR protein that catalyzes DXP to produce MEP.

### 3.4. SaDXR Is Localized in Chloroplasts

TMHMM version 2.0 prediction software showed that SaDXR had no transmembrane domain (Appendix A). According to the results of ChloroP 1.1 (http://www.cbs.dtu.dk/services/ChloroP/, accessed on 15 April 2019) analysis, the SaDXR sequence contains an N-terminal chloroplast transit peptide (cTP) in the 50 amino acid regions. Almost all plant MEP pathway genes have the N-terminal transit peptide sequence, which transports proteins into plastids [6]. To further analyze the localization of SaDXR, the full-length SaDXR in frame to YFP was transiently transformed into the mesophyll protoplasts of A. thaliana seedlings. As expected, yellow fluorescence of SaDXR-YFP coincided exactly with chlorophyll autofluorescence (Figure 4), suggesting that SaDXR was targeted in chloroplasts. This result was consistent with previous research on the localization of DXRs [6,8] that indicated that the SaDXR protein may be involved in plastidial isoprenoid biosynthesis in plants.

### 3.5. Isolation and Characterization of the SaDXR Promoter

An 897-bp DNA fragment upstream to the ATG of the DXR gene (accession no. MH018576) was cloned from gDNA of *S. album* through FPNI-PCR [23,24], and specific PCR amplification used primers listed in Appendix A. The obtained full nucleotide sequence is shown in Figure 5. Using the TSSP program, the putative TSS of the cloned promoter was predicted and was shown to be located 164 bp upstream of the SaDXR ATG start codon. Using PlantCARE software, a well-conserved core promoter cis-element TATA-box (TAATA) was predicted 22 bp upstream of TSS. Another common cis-acting element, the CAAT-box (CAAAT) for RNA polymerase binding in the promoter, was labeled at −210 to −197. Other putative cis-acting regulatory elements were also identified (Figure 5). Among them, some are involved in hormone regulation, such as the abscisic acid-responsive element (ABRE), the gibberellin-responsive element (GARE-motif and TATC-box), and the salicylic acid-responsive element (TCA-element). Moreover, light-responsive elements, including the ATCT-motif, G-box, GT1-motif, Sp1, and chs-CMA2a, were found. The G-box generally binds with basic leucine zipper (bZIP) transcription factors, especially MYC2, which plays a critical role in the synthesis of secondary metabolites [34,35,36]. Notably, a heat stress-responsive cis-element (HSE), an element related to the regulation of zein metabolism (O2-site), a cis-element required for endosperm expression (Skn-1_motif), and a cis-acting element involved in defense and stress-responsive elements (TC-rich repeats), were found in the SaDXR promoter (Figure 5). These results indicate that the SaDXR promoter contains various cis-acting regulatory elements, and further suggests that the transcriptional activity of the SaDXR promoter is complex, and might be controlled by different environmental signals.

### 3.6. Tissue-Specific Expression of SaDXR

RNA was isolated from different tissues (roots, twigs, stem sapwood, leaves, flowers, fruit and stem heartwood) of 7-year-old *S. album* trees. The expression pattern and level of SaDXR were monitored by qRT-PCR. Although SaDXR was detected in all tissues, it displayed tissue-specific expression, exhibiting a higher level of expression in organs with photosynthetic pigments, including twigs, leaves and flowers (Figure 6). The highly valuable aromatic heartwood or essential oil, which contains sandalwood-specific terpenoids (i.e., santalol), form in the stems and roots of adult trees that are at least five years old. Additionally, terpenoids are biosynthesized via MEP and MVA pathways, and DXR is the critical rate-limiting enzyme in the MEP pathway [6]. Thus, this tissue expression analysis also determined whether SaDXR expression was consistent with the content of sandalwood-specific terpenoids in different tissues of *S. album*. However, the expression of SaDXR was low and did not match the expression patterns of a gene critical for the biosynthesis of terpenoids, *S. album* santalene synthase (SaSSy), which was highly expressed in tissues in which sandal terpenoids are typically biosynthesized, i.e., roots and stem heartwood (Figure 6).

### 3.7. Expression of SaDXR in Response to Light

In various species, DXRs promote the biosynthesis of plastidial isoprenoids, especially the photosynthetic pigments carotenoids and chlorophyll, and light is an important regulatory factor in biosynthesis of photosynthetic pigments [36,37,38,39]. A number of light-responsive cis-elements were identified in the SaDXR promoter. To study whether light regulates SaDXR expression, 40-day-old *S. album* etiolated seedlings were grown under constant darkness and then exposed to light. Light exposure for 1 or 3 h induced only a minor change in the expression level of SaDXR, peaking at 6 h of illumination (Figure 7).

### 3.8. Chlorophylls and Carotenoids Increased in 35S::SaDXR Overexpression Arabidopsis Lines

To test whether SaDXR regulates the biosynthesis of plastidial isoprenoids, an overexpression vector 35S::SaDXR was constructed (Figure 8A) and transformed into A. thaliana using an A. tumefaciens-mediated method. Kanamycin-resistant T2 plants were further identified by semi-quantitative RT-PCR. SaDXR transcription was detected in all four overexpression lines (L1, L3, L5, L6) but not in the WT plants. Additionally, L3 had the highest expression level compared to other overexpression lines (Figure 8B,C). All plants, including transgenic and WT plants, had similar expression levels of AtDXR, suggesting that SaDXR was successfully transcribed and expressed in A. thaliana. Compared to WT plants, chlorophyll content, including chlorophyll a, b and total chlorophyll, were significantly higher in the four overexpression lines. In L3, chlorophyll a, b, total chlorophyll and carotenoid content increased 16.0, 24.8, 18.2 and 11.8%, respectively, more than in WT plants, which only contained 1.0733, 0.3739, 1.4472 and 0.1888 mg g^−1^ FW, respectively (Figure 9). Apart from L5, other overexpression lines had a higher carotenoid content than WT plants. These results suggest that SaDXR derived from the MEP pathway was involved in the biosynthesis of plastidial isoprenoids. However, none of the overexpression lines displayed phenotypes different from WT plants.

### 3.9. Transcript Profiles of SaDXR under MeJA, H_2_O_2_

Generally, only the stems and roots of adult sandalwood (at least five years old) produce aromatic heartwood or essential oil, with sandalwood terpenoids [40]. Recently, some studies suggested that the contents of sandalwood sesquiterpenoids, such as α-santalol and β-santalol, were increased in the stems of young sandalwood trees that were treated by MeJA and H_2_O_2_ [3,41,42]. To test whether the expression pattern of SaDXR is regulated by these elicitors, the roots of sandalwood seedling were treated with MeJA and H_2_O_2_, and gene expression was examined by qRT-PCR. SaDXR expression gradually increased in the H_2_O_2_ treatment and peaked at 6 h, attaining an approximately 4-fold higher expression level, followed by a gradual decline. Finally, SaDXR expression level returned to its original level (Figure 10). These expression profiles were quite consistent with the expression patterns of SaSSy. Compared to H_2_O_2_ treatment, MeJA treatment had less effect on the expression of SaDXR, which increased at 6 h and 9 h, and the expression at other times was similar to those in the control. This result was similar to the expression of SaSSy, which was also enhanced at 6 h and 9 h, but then continuously maintained a high expression level at 12 h and 24 h (Figure 10).

### 3.10. Metabolite Analysis in the FOS Treatment

To investigate further effects of SaDXR on sandalwood sesquiterpenoid biosynthesis, the DXR enzyme was inhibited and the MEP pathway flux was blocked. Tissue samples from seedlings roots were collected at 7 days after 20 µM FOS employ. Almost consistent with previous studies [17,41], the main sandalwood sesquiterpenoids, namely α-santalene, α-bergamotene, Epi-β-santalenem, β-santalene and α-santalol, were detected in the control and FOS-treated seedlings roots (Figure 11 and Appendix A). The metabolite analysis showed that FOS treatment decreased the content of sandalwood sesquiterpenoids, especially α-santalol, which is the most essential ingredient, significantly reduced from 2.0444 µg g^−1^ to 1.1288 µg g^−1^ between the control and FOS-treated roots. Compared to the control, total detected sandalwood sesquiterpenoids were also decreased by 24.34% in FOS treatment (Figure 11).

## 4. Discussion

Sandalwood sesquiterpenoids are the main ingredients for high value essential oil while photosynthetic pigments are essential for the growth and development of sandalwood plants. DXRs are critical enzymes that catalyze the secondary rate-limiting step in the MEP pathway, which plays a vital role in the biosynthesis of plastidial isoprenoids, including growth hormones and photosynthetic pigments, and provides flux to various terpenes. In the MEP pathway, apart from the DXS enzyme, which is encoded by multiple gene families, other enzymes exist as a single-copy in most plant species. There are only two-copy DXRs in *Hevea brasiliensis* and *Glycine max* [38,43], while almost all other plant species have a single copy (Appendix A). In this study, we cloned and functionally analyzed a single *SaDXR* gene from *S. album*. Amino acid sequence alignment, as well as 2D and 3D structures of SaDXR indicated that it likely belonged to the superfamily of DXR enzymes and may have the same catalytic function as other DXR proteins. There are two characteristic domains in this superfamily, a conserved NADPH binding site (GSTGSIGT) and a transit peptide for the plastid motif Cys-Ser-Ala/Met/Val/Thr. At the genomic level, a detailed study of genomic *SaDXR*, *AtDXR*, *PtDXR* and *VvDXR* suggested that *SaDXR* also has a conserved genomic organization as other plant *DXR* genes. Furthermore, a phylogenetic tree analysis suggested strong evolutionary conservation of the *DXR* superfamily. Supported by proteomics and in silico studies, almost all MEP-pathway enzymes in plant species have an N-terminal cTP, which enables the transport of proteins into plastids [8,9]. *SaDXR* contains an N-terminal cTP, assessed by analysis with the ChloroP 1.1 program, and the result of subcellular localization suggested that the SaDXR protein was targeted to chloroplasts where plastid isoprenoids may be biosynthesized.

Previous studies on the molecular function of *DXR* genes mainly focused on their involvement in the biosynthesis of photosynthetic pigments, terpenoids and their derivatives (Appendix A). The result of overexpression analyses in model plants and the *DXR* mutant revealed that *AtDXR* [44,45], *SyDXR* [46], *AvDXR* [37] and *GmDXR1/2* [38] collectively played a critical role in the biosynthesis of photosynthetic pigments (Appendix A). Additionally, experiments using a transgenic system, RNAi, and the inhibitor FOS indicated that *DXR* genes also shift flux toward different terpenoids and their derivatives such as forskolin [47], taxadiene [44], artemisinin [48,49], tanshinone [50,51,52], aethiopinone [53], lanosterol/cholesterol [54] and triptolide/cerlastrol [55]. In order to understand the relationship between *SaDXR* and photosynthetic pigments and sandalwood sesquiterpenoids, this study analyzed the expression pattern of *SaDXR* in different tissues. Tissues with photosynthetic pigments, namely twigs, leaves and flowers, had higher expression levels of *SaDXR* than other tissues. In addition, light plays a major role in the accumulation of photosynthetic pigments [56] and regulation of *SaDXR* expression in the etiolated seedlings of *S. album*. Our results indicate that *SaDXR* expression is consistent with previous studies that show the relevance of the *DXR* gene to photosynthetic pigments. Furthermore, transgenic *Arabidopsis* lines verified that the increased expression of *SaDXR* resulted in a higher accumulation of chlorophylls and carotenoids.

Sandalwood sesquiterpenoids are exclusively biosynthesized in stem heartwood and roots in adult sandalwood trees. However, *SaDXR* expression level was not consistent with the content of sandalwood sesquiterpenoids and the expression patterns of *SaSSy* in different tissues. The result was not a surprise because *SaDXR* might only play a minor role in the biosynthesis of sandalwood sesquiterpenoids. Additionally, the production of many secondary metabolites, including sandalwood sesquiterpenoids, were increased by external stimuli [27,57,58,59]. Based on the use of MeJA and H_2_O_2_ to increase the content of sandalwood sesquiterpenoids [41,42], this study noted that *SaDXR* expression was affected by MeJA and H_2_O_2_ treatment, showing a similar expression pattern as *SaSSy*. The overexpression of *SaDXR* in *Arabidopsis* or tobacco is invalid because these are not enzymes of biosynthesis of sandalwood sesquiterpenoids in these model plants. Similar to other studies [48,51,54,60], a specific inhibitor for *DXR* was tested. The key component of sandalwood sesquiterpenoids, α-santalol, decreased in the FOS treatment. These results suggest that *SaDXR* controls the flux towards sandalwood sesquiterpenoids, although its contribution may be minor. This study cannot fully illustrate the role of *SaDXR* in the biosynthesis of sandalwood sesquiterpenoids. Thus, in order to further confirm the function of *SaDXR*, the establishment of a transgenic system for sandalwood is essential. Recently, our team has been making a concerted effort to establish a system for transgenic sandalwood with an emphasis on hairy roots, callus and cell suspensions. Not only will the function of *SaDXR* be adequately expounded, so too will a deeper understanding of the molecular mechanism of sandalwood sesquiterpenoids emerge in such transgenic systems.

Many regulatory proteins, i.e., transcription factors (TFs), recognize specific *cis*-elements in the promoters of target genes and activate or repress the expression of genes, and finally regulate the accumulation of isoprenoids, especially secondary metabolites in a spatial and temporal manner [61]. Compared to a single gene, TFs regulate the biosynthesis of isoprenoids more efficiently because they usually regulate the transcription of a series of genes in a specific pathway [62]. In the present study, the promoter region of *SaDXR* was cloned and *cis*-elements were analyzed. It is worth noting that the three G-boxes found in *SaDXR* promoter are primary involved in light responsiveness but also play a critical role in MeJA responsiveness. The basic helix-loop-helix (bHLH) TF MYC2 binds to G-boxes within the promoter of various genes, including the *Nicotiana tabacum* putrescine N-methyltransferase gene (*NtPMT*) [63], the *Artemisia annua* cytochrome P450 monooxygenase (AaCYP71AV1)/ artemisinic aldehyde Δ11 (13) reductase (*AaDBR2*) gene [36] and the *Aquilaria sinensis* δ-guaiene synthase (*AsASS1*) gene [64], to activate MeJA-induced transcription [65,66], finally resulting in an increase in the content of nicotine, artemisinin and δ-guaiene, respectively. Therefore, there also may be specific MYC2 TFs that bind to the *SaDXR* promoter and participate in the regulation of biosynthesis of sandalwood terpenoids when stimulated by MeJA.

## 5. Conclusions

The *SaDXR* gene was successfully isolated in *S. album* based on a transcriptome database. Bioinformatics and phylogenetic analyses suggest that *SaDXR* belongs to the conservative *DXR* gene family. Additionally, the *SaDXR* promoter was cloned and *cis*-elements were analyzed. The ectopic overexpression of *SaDXR* in Arabidopsis, combined with the expression pattern analysis of tissue and light treatment, demonstrated that the *SaDXR* gene is involved in the biosynthesis of chlorophylls and carotenoids. Moreover, *SaDXR* expression in MeJA and H_2_O_2_ treatments and metabolite analyses in the FOS treatment indicated that *SaDXR* enhances flux to biosynthesis of sandalwood sesquiterpenoids. These findings provide a more solid basis for understanding the genetic regulation of the biosynthesis of sandalwood sesquiterpenoids, eventually allowing, in future research, the biotechnological improvement of sandalwood.

## Figures and Tables

**Figure 1 genes-12-00626-f001:**
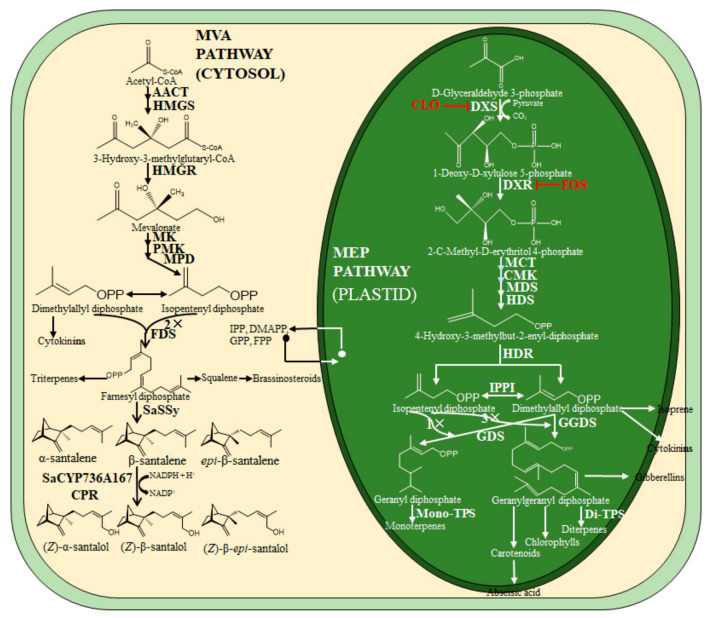
Outline of the MVA and MEP pathways in plants, and the biosynthesis of santalol, chlorophylls and carotenoids in *S. album*. DXR was inhibited by fosmidomycin (FOS), which is shown in red. Cytosol: AACT, acetoacetyl-CoA thiolase; FDS, farnesyl diphospate synthase; HMGR, 3-hydroxy-3-methylglutary-CoA reductase; HMGS, 3-hydroxy-3-methylglutary-CoA synthase; MK, mevalonate kinase; MPD, mevalonate-5-diphosphate decarboxylase; MVA, mevalonate acid; PMK 5-phosphate-mevalonate kinase. Plastid: CMK, 4-cytidine 5′-diphospho-2-C-methyl-D-erythritol kinase; DMAPP, dimethylallyl diphosphate; DXP, 1-deoxy-D-xylulose-5-phosphate; DXR, 1-deoxy-D-xylulose-5-phosphate reductase; DXS, 1-deoxy-D-xylulose-5-phosphate synthase; GDS, geranyl diphosphate synthase; GGDS, geranylgeranyl diphosphate synthase; GGPP, geranylgeranyl diphosphate; GPP, geranyl diphosphate; HDR, 1-hydroxy-2-methyl-2-(E)-butenyl-4-diphosphate reductase; HDS, 1-hydroxy-2-methyl-2-(E)-butenyl-4-diphosphate synthase; IPPI, isopentenyl diphosphate isomerase; MCT, 2-C-methyl-D-erythritol 4-phosphate cytidylyltransferase; MEP, 2-C-methyl-D-erythritol-4-phosphate.

**Figure 2 genes-12-00626-f002:**
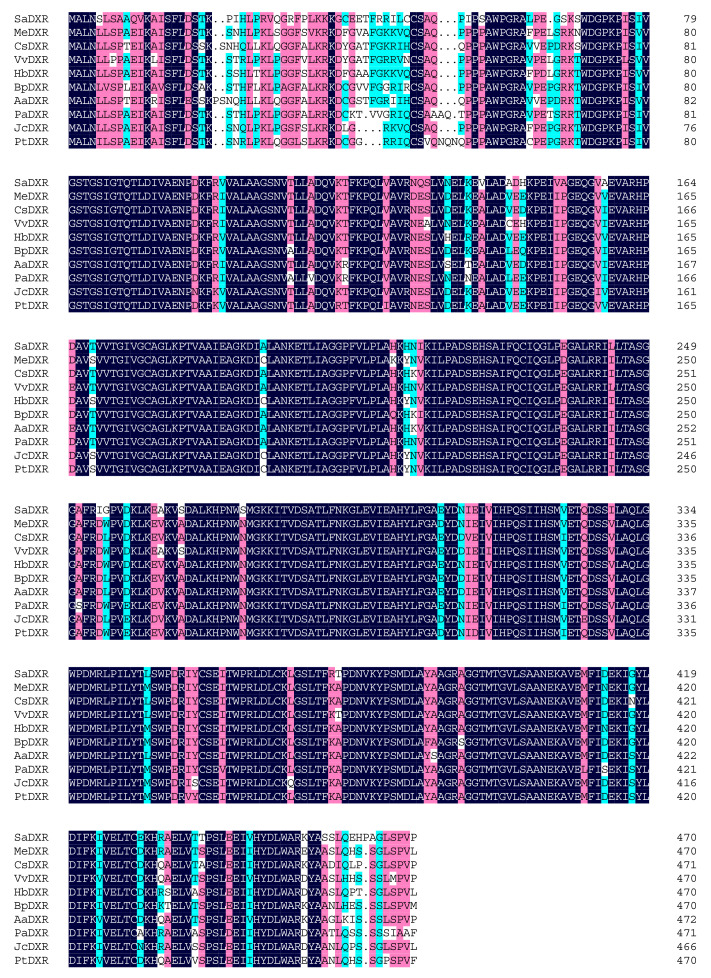
Alignment of deduced amino acid sequences of SaDXR, and related proteins from AaDXR (*Actinidia arguta*, AID55340.1), BpDXR (*Betula platyphylla*, AHX36947.1), CsDXR (*Camellia sinensis*, AKE33276.1), HbDXR (*Hevea brasiliensis*, XP_021669849.1), JcDXR (*Jatropha curcas*, NP_001295720.1), MeDXR (*Manihot esculenta*, XP_021616674.1), PaDXR (*Prunus avium*, XP_021826007.1), PtDXR (*Populus trichocarpa*, XP_002318048.2), and VvDXR (*Vitis vinifera*, XP_002282761.1). The conserved NADPH binding motif (GSTGSIGT) and Cys-Ser-Ala motif are indicated by a horizontal line and asterisk, respectively. Alignments were performed with DNAMAN version 7. Black: 100% identity; pink: 75% ≤ identity < 100%; blue: 50% ≤ identity ≤ 75%.

**Figure 3 genes-12-00626-f003:**
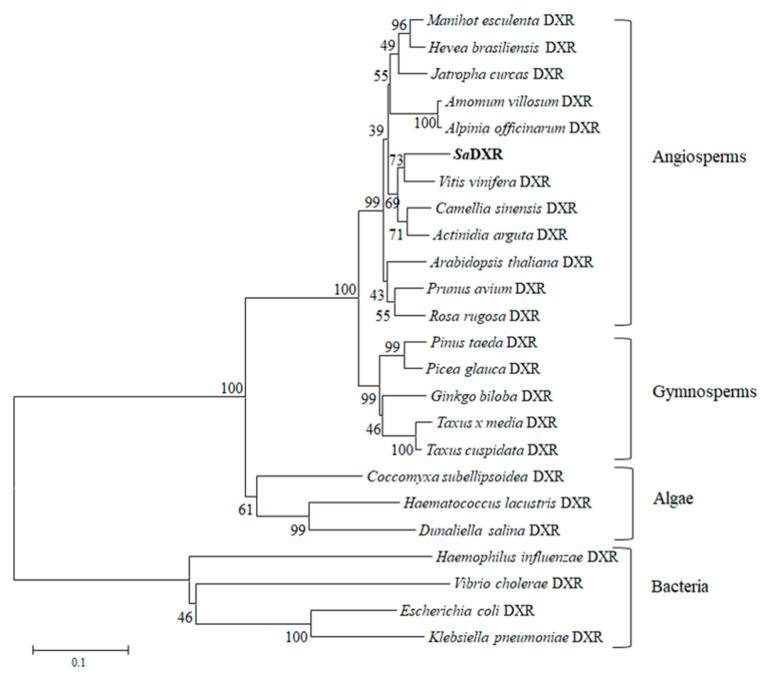
Phylogenetic analysis of the amino acid sequences of DXRs. The neighbor-joining phylogenetic tree was constructed using the bootstrap method with MEGA 5.1 (1000 bootstrap replications). The protein sequences used in this tree are: *Alpinia officinarum* (AEK69520.1), *Amomum villosum* (ACS26204), *Arabidopsis thaliana* (NP_201085), *Actinidia arguta* (AID55340.1), *Camellia sinensis* (AKE33276.1), *Coccomyxa subellipsoidea* (EIE23728.1), *Dunaliella salina* (ACT21081.1), *Escherichia coli* (NP_414715), *Ginkgo biloba* (AAR95700), *Haematococcus pluvialis* (AEY80027.1), *Haemophilus influenza* (NP_438967.1), *Hevea brasiliensis* (ABD927022), *Jatropha curcas* (NP_001295720.1), *Klebsiella pneumoniae* (CDO15900.1), *Manihot esculenta* (XP_021616674.1), *Picea glauca* (JAI17646.1), *Pinus taeda* (ACJ67022 ), *Prunus avium* (XP_021826007.1), *Rosa rugosa* (AEZ53171), *Taxus cuspidata* (AAT47184.1), *Taxus × media* (AAU87836), *Vibrio cholerae* (NP_231885), and *Vitis vinifera* (XP_002282761.1).

**Figure 4 genes-12-00626-f004:**
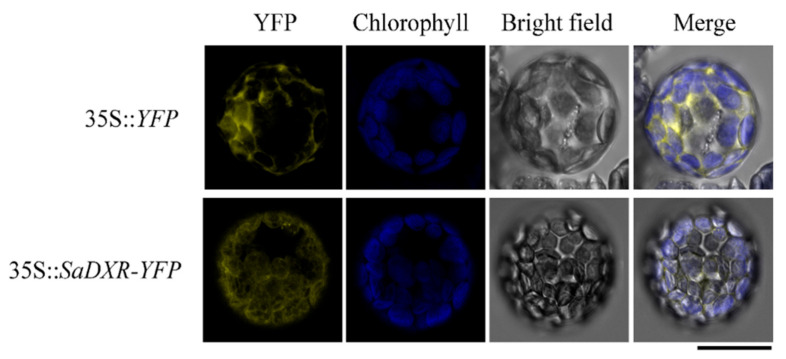
Subcellular localization of SaDXR. The signal was observed by a confocal laser scanning microscope. Transient expression of 35S::*YFP* and 35S::*SaDXR-YFP* in living protoplasts, displaying YFP fluorescence (yellow), chlorophyll autofluorescence (blue), bright field, and merged image (yellow plus blue). Bar: 20 μm.

**Figure 5 genes-12-00626-f005:**
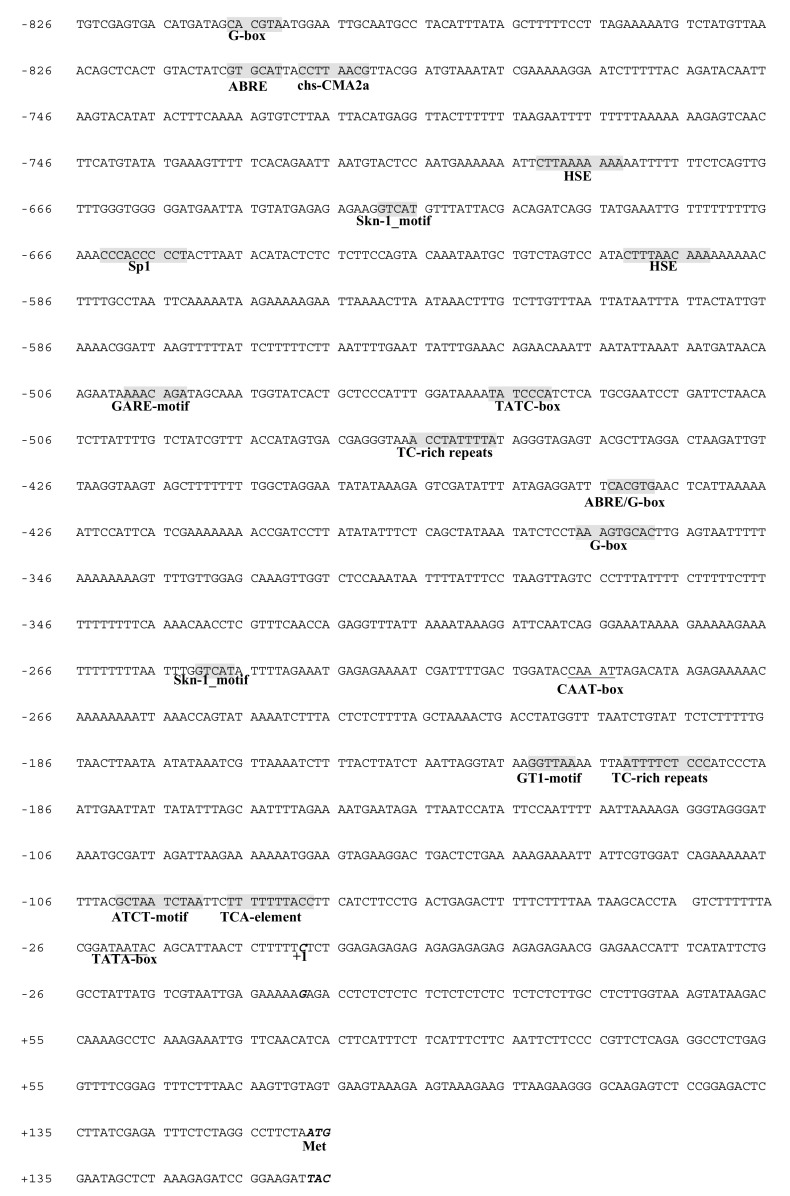
Nucleotide sequence and putative cis-acting elements of the cloned SaDXR promoter. The transcription initiation site and ATG are shown in bold italics, and indicated as +1 and +164, respectively. The TATA and CAAT-box are underlined and labeled from −22 to −18 and from −210 to −197, respectively. Various cis-acting elements were predicted by the PlantCARE program. ABRE, a cis-acting element involved in abscisic acid responsiveness; ATCT-motif, G-box, GT1-motif, Sp1, and chs-CMA2a are light-responsive elements; GARE-motif and TATC-box are cis-acting elements involved in gibberellin responsiveness; HSE, a cis-acting element related to responsiveness to heat stress; O2-site is involved in the regulation of zein metabolism; Skn-1_motif, a cis-acting regulatory element required for endosperm expression; TC-rich repeats, cis-acting elements involved in defense and stress responsiveness; TCA-element, a cis-acting element involved in salicylic acid responsiveness.

**Figure 6 genes-12-00626-f006:**
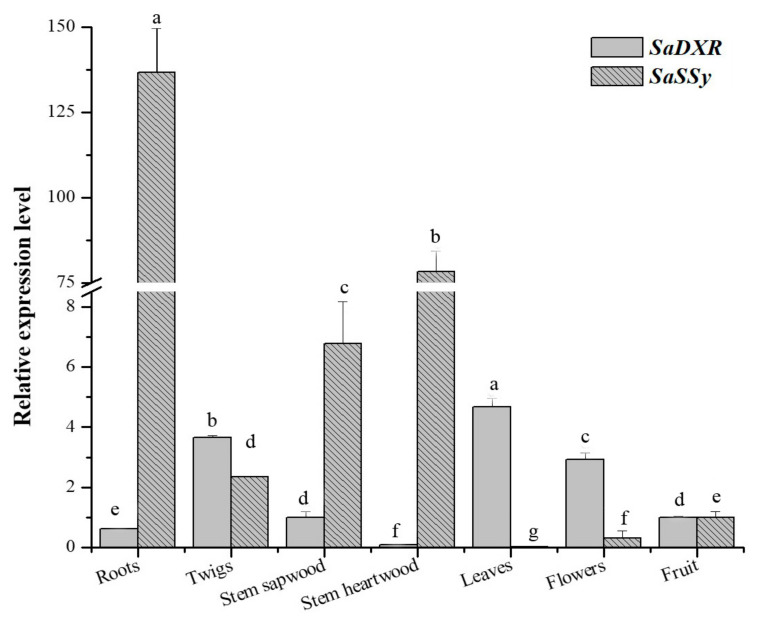
Expression patterns of SaDXR and SaSSy in different organs of a 7-year-old S. album trees analyzed by qRT-PCR. Each value represents the means of three independent biological replicates ± SD. Different letters (a,b,c,d,e,f,g) above bars indicate significant differences among organs, assessed separately for SaDXR and SaSSy (DMRT, *p* < 0.05).

**Figure 7 genes-12-00626-f007:**
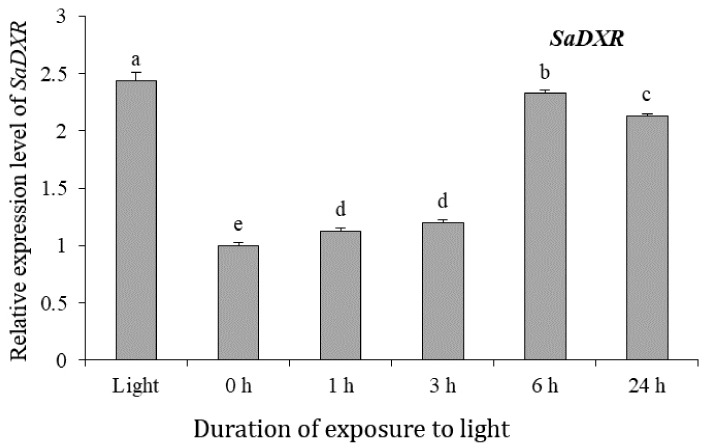
Expression profile of SaDXR in 40-day-old S. album etiolated seedlings that were exposed to light. S. album seeds were germinated in the dark for 20 days then grown into 10 cm-etiolated seedlings for other 20 days. Subsequently, etiolated seedlings were exposed to light for 0, 1, 3, 6, 24 h or grown in a 16-h photoperiod. Each value represents the means of three independent biological replicates ± SD. Different letters above bars indicate significant differences (DMRT, *p* < 0.05).

**Figure 8 genes-12-00626-f008:**
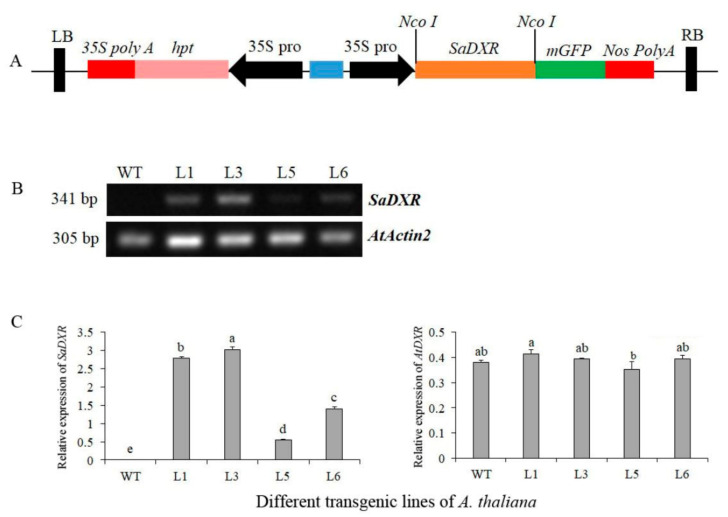
Overexpression of SaDXR in A. thaliana. (**A**) Schematic representation of the 35S::SaDXR overexpression vector. (**B**) Gene expression analysis of SaDXR in wild-type (WT), i.e., control, and in select transgenic lines of A. thaliana by using semi-quantitative PCR. (**C**) Expression level of SaDXR and AtDXR in WT or transgenic A. thaliana seedlings by qRT-PCR. WT did not give a signal as the primers are specific to SaDXR. Each value represents the means of three independent biological replicates ± SD. Different letters above bars indicate significant differences (DMRT, *p* < 0.05).

**Figure 9 genes-12-00626-f009:**
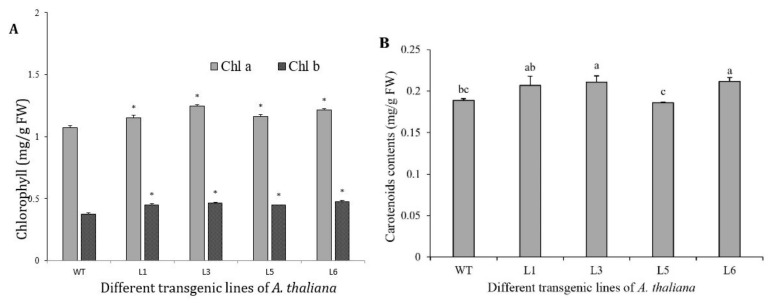
Contents of chlorophylls and carotenoids in transgenic *A. thaliana* seedlings carrying the *SaDXR* transgene. (**A**) Contents of chlorophyll *a* (Chl *a*), chlorophyll *b* (Chl *b*) and total chlorophyll in transgenic *A. thaliana* plants. (**B**) Content of carotenoids in *35S::SaDXR* transgenic lines. Each value represents the means of three independent biological replicates ± SD. Different letters above bars indicate significant differences for each Chl type relative to WT (DMRT, *p* < 0.05).

**Figure 10 genes-12-00626-f010:**
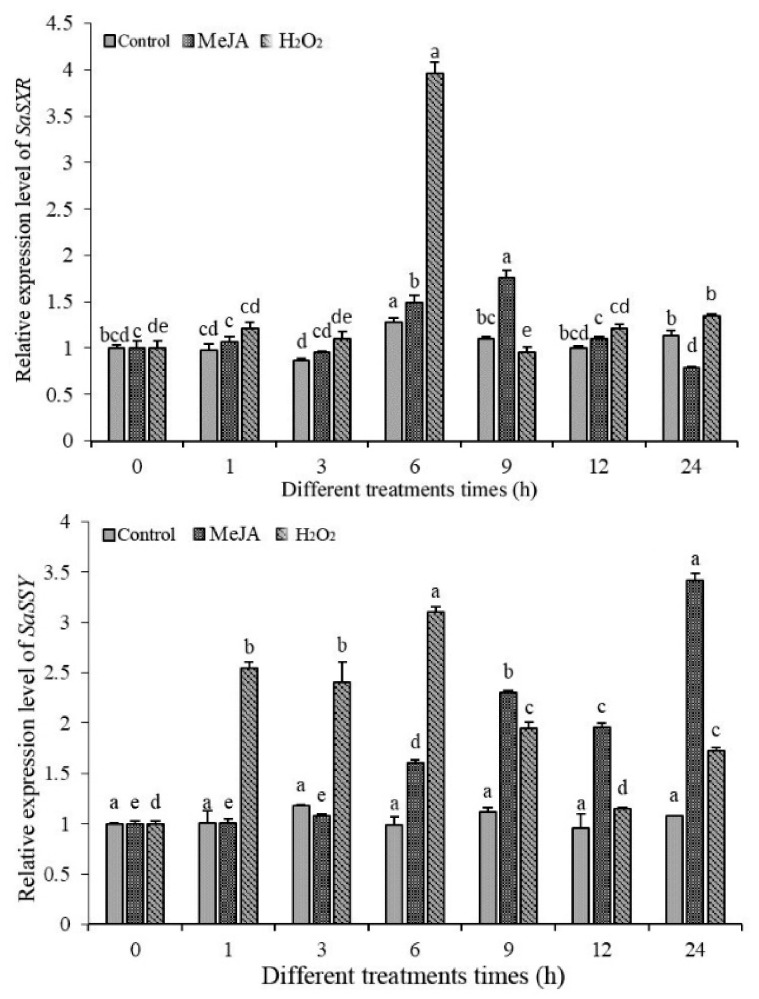
Expression analysis of SaDXR and SaSSy of 8-month-old *S. album* seedlings in response to two elicitors, 5 mM H_2_O_2_ or 1 mM MeJA. Eight-month-old *S. album* seedlings treated with water were used as the control. Each value represents the means of three independent biological replicates ± SD. Different letters (a,b,c,d,e,f,g) above bars indicate significant differences across treatments (DMRT, *p* < 0.05).

**Figure 11 genes-12-00626-f011:**
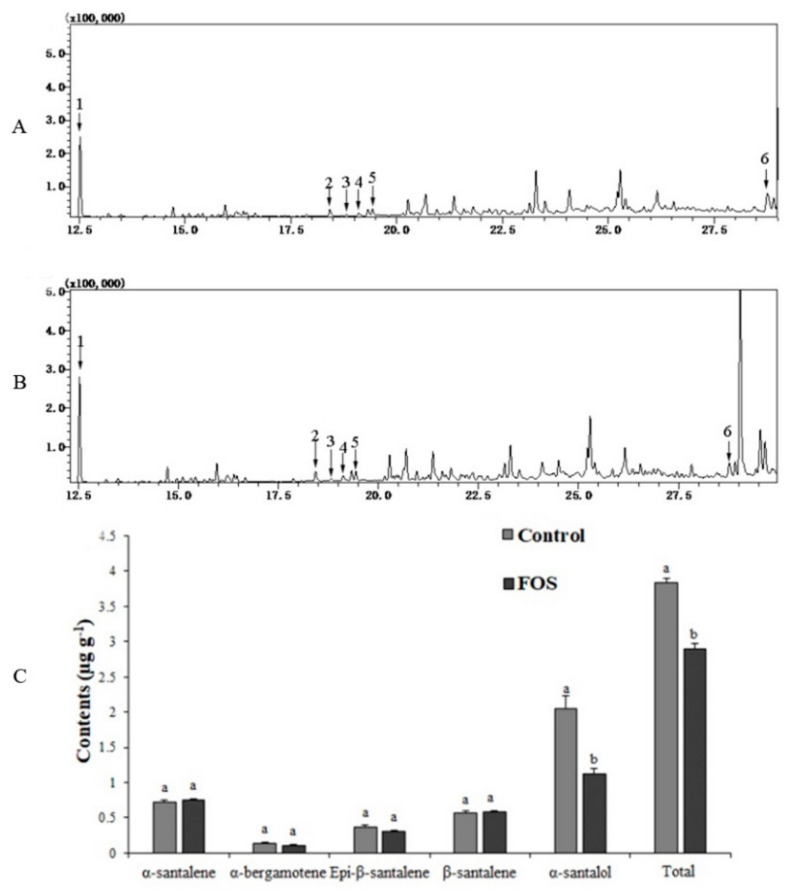
Profile of sandalwood sesquiterpenoids in the roots of the control and FOS-treated 8-month-old seedlings. (**A**,**B**) Total ion chromatogram (TIC) of control and FOS-treated seedlings roots, respectively. (**C**) Relative contents of sandalwood sesquiterpenoids (1, dodecane; 2, α-santalene; 3, α-bergamotene; 4, epi-β-santalene; 5, β-santalene; 6, α-santalol) were identified by comparison to NIST05, NIST05s and the Wiley 9 library. Each value represents the means of three independent biological replicates ± SD. Different letters above bars indicate significant differences between FOS and the control for each sesquiterpenoid (DMRT, *p* < 0.05).

## Data Availability

The study did not report any other data.

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
