# Peer review of "Molecular Cloning and Functional Analysis of 1-Deoxy-D-Xylulose 5-Phosphate Reductoisomerase from Santalum album"

_genes, 2021, doi:10.3390/genes12050626_

Round 1

Reviewer 1 Report

The manuscript of Zhang et al. reports the molecular characterization of 1-deoxy-D-xylulose 5-phosphate reductoisomerase (DXR) gene of Santalum album. The authors cloned the gene from Santalum album, reported their expression profiles in different plant parts and under different treatments of MeJa, H2O2 and light, and functionally characterized the gene involved in biosynthesis of photosynthetic pigments by overexpressing the gene in heterologous system. Even though the gene is already characterized in other plant species and the novelty of the study is limited to only Santalum album, I think it’s a good addition to the literature. However, there are a number of issues that must be addressed prior to publication. Some of the comments or suggestions are given below-

General comments:

- The whole manuscript is poorly written and needs a major improvement in English. Some of the statements are missing words or have wrong words. Some sentences are grammatically incorrect right from abstract to all other sections, some sentences even do not make any sense and need to be rephrased (please see the attached manuscripts for the marked sentences and suggestions). The flow of reading is missing in some sections of the manuscripts. I would suggest to improve or edit the English of the whole manuscript by a native English speaker or through some editing service to improve the flow of reading and make the study more understandable to the readers.

- The title ‘Molecular cloning and functional identification’ does not reflect the content of the manuscript. The words “functional identification” is also confusing, authors may think about changing the title.

Specific comments:

-In the result section, the title of the subsection 3.2 should be ‘Analysis of SaDXR gene’ NOT the ‘Analysis of genomic DNA of SaDXR’. Besides, the authors started the subsection with the protein information which does not make any sense here. Please put that protein information in the appropriate place.

L27: The word ‘conservative’ is not right word in this sentence. It should be ‘the SaDXR encodes a functional and highly conserved DXR protein’. The word ‘conservative’ has also been wrongly used in L449 sentence

-All the gene and transcripts name should be written in italic form.

-Please edit or rephrase these sentences to make them meaningful and understandable to the readers-

L45-46, L62-63, L102-103, L105, L111-113, L117-118, 120, L153-L154, L168-169, L175-176, L181, L183, L186-188, L212-214, L263-265, L268-269, L292-295, L342-344, L358-359, L373, L375-376, L382, L442-443, L452, L454

-L136 (section 2.3), L257 (section 3.3)-The ‘Bioinformatics analysis’ may not be the right words, please replace them with right words.

-L155: I would suggest the authors to change the subtitle to ‘Subcellular localization of SaDXR’

-L175: Put the references for delta-delta method (RT-PCR)

-L335: Please change the subtitle of the section 3.6

-Please elaborate any abbreviated word when used for the first time (e.g., SOPMA in L143).

-L458: Not sure what authors have meant by ‘pattern plants’ in this sentence.

-L463-464: Please combine these two sentences to make it meaningful.

-I don’t see any information about data availability.

Fig 1: Please replace the word ‘Plasmid’ with ‘Plastid’ (see attached manuscript)

Fig 7, 8(C-D): Please put the title in the x axis of the figures.

Fig 8(C-D): Please correct the title of the y axis, c) Relative expression of SaDXR and d) Relative expression of atDXR.

Fig 9: Missing the legend for ‘Total chlorophyll’ (A) and title of the x axis (A, B). Please also correct the title of the x axis to a) Chlorophyll content (…) instead of Chlorophyll, and b) Carotenoid content (…) instead of Carotenoid.

Fig 10: Missing the title of the x axis (A, B). Please also correct the title of the y axis like Fig 8 (C, D) as mentioned above. I would also suggest authors to use the word ‘Control” instead of ‘CK’. If the abbreviation CK is used, it should be explained in the figure caption.

Fig S2: Please write more details about left and right panels of the figure in the caption to make it more understandable.

Fig S4: Please put more details of the figure in the caption.

Fig S5: The table S3 is redundant and can be merged with Fig S5 just by putting the intron size in the figure.

Fig S6: Very confusing, which mass spectra is for control and which one for FOS treatment. Please put more details to make it understandable.

Please also look carefully at the attached manuscript to see the reviewer’s comments and suggestions and other corrections/typos that were not mentioned here.

I would suggest accepting the paper with major revision. I strongly encourage the authors to improve the English of the whole manuscript by a native English speaker or through some editing service, improve the overall flow of reading of the manuscript, put enough information to make the titles/captions of the figures self-explanatory and resubmit a REVISED manuscript.

Author Response

We have answer all the questions. Please see the attched file.

We have also asked our coauther Dr. Jaime to improve the language.

Reviewer 2 Report

In the submitted manuscript “Molecular cloning and functional identification of 1-deoxy-D-xylulose 5-phosphate reductoisomerase from Santalum album” Zhang et al. attempted to clone the 1-deoxy-D-xylulose-5-phosphate reductoisomerases (DXRs) gene in S. album using transcriptome datasets and detected the expression patterns of SaDXRgene in various tissues (root, twig, stem sapwood, leaf, flower, fruit and stem heartwood) under different treatments (MeJA, H2O2and light). They found the relationship between SaDXR and chlorophylls, carotenoids, sandalwood sesquiterpenoids biosynthesis using the overexpression lines of Arabidopsis thaliana and DXRinhibitor (FOS) treatment. This is a good report with the experiments reasonably well arranged to convey the key messages. The conclusions are well supported by the data presented and some interesting results presented in the manuscript will provide a better understanding of molecular mechanisms of SaDXRbiosynthesis of photosynthetic pigments and shift flux to sandalwood sesquiterpenoids.

Minor point:

Some parts of manuscript writing require changes for both clarity and proper English usage.

Author Response

Thank you so much for your positive suggestion and we have revised the manscript according to your suggestion

Round 2

Reviewer 1 Report

The authors addressed almost all the issues/concerns I raised, and the revised version has been improved a lot and has now nice flow of reading.

Having said that there are still some typos that need to be corrected. For instances, I can still see the word “Plasmid” has not been changed in Fig 1 which should be replaced by the word “Plastid”. Same with the Figure 1 caption. In L168, ‘stored’ should be replaced by ‘frozen’. Even though authors said that they merged Table S3 with Fig S5 (as I suggested by putting the intron size (base pair) in the Fig S5), but I still see the separate Table S3 and Fig S5 in the supplementary materials. I also want to mention that it is very difficult to read and track the sentences in the manuscript with so many corrections in the text. I believe the Editor/English editor will take care of that. 

Author Response

We have answered all the questions.
